# Monitoring the Spread of Grapevine Viruses in Vineyards of Contrasting Agronomic Practices: A Metagenomic Investigation

**DOI:** 10.3390/biology12101279

**Published:** 2023-09-25

**Authors:** Minh Vu, Wendy McFadden-Smith, Sudarsana Poojari

**Affiliations:** 1Cool Climate Oenology and Viticulture Institute, Brock University, St. Catharines, ON L2S 3A1, Canada; 2Ontario Ministry of Agriculture, Food and Rural Affairs, Lincoln, ON L0R 2E0, Canada

**Keywords:** sentinel vine, HTS, PCR, grapevine viruses, grapevine red blotch virus, grapevine pinot gris virus, alternative hosts, secondary infection

## Abstract

**Simple Summary:**

Grapevine red blotch virus (GRBV) and grapevine Pinot gris virus (GPGV) are two of the most concerning grapevine viruses in Canada. Currently, the secondary spreading of these viruses (spreading via an insect vector) in Canada is poorly documented. To monitor the spreading of these viruses, we deployed virus-free grapevines onto two virus-infected vineyards with different agronomic practices. One vineyard was organic while the other was conventional. Using two independent diagnostic tools, namely the polymerase chain reaction (PCR) and the high-throughput sequencing technique (HTS), we have found that GPGV is spreading in both vineyards. The spread of GPGV was noticeably faster in the organic vineyard than in the conventional one, possibly due to a higher insect population. There was no evidence of the spreading of GRBV. A small collection of plant species in and around the vineyard was also tested by the two methods above. None of the non-grape species harbored any grapevine viruses but a large portion of the wild grape found on the edge of the vineyard carried GPGV. This study is the first direct proof of insect-mediated spreading of GPGV in North America.

**Abstract:**

This study investigated the transmission of grapevine viruses, specifically grapevine red blotch virus (GRBV) and grapevine Pinot gris virus (GPGV), in vineyards in Niagara Region, Ontario, Canada. Forty sentinel vines that were confirmed free of GRBV and GPGV by both high-throughput sequencing (HTS) and endpoint polymerase chain reaction (PCR) were introduced to two vineyards (one organic and one conventional) that were heavily infected with both GRBV and GPGV. Four months post-introduction, the sentinel vines were relocated to a phytotron. The HTS results from 15 months post-introduction revealed a widespread infection of GPGV among the sentinel vines but did not detect any GRBV. The GPGV infection rate of sentinel vines in the organic vineyard (13/18) was higher than in the conventional vineyard (1/19). The possibility of an alternative viral reservoir was assessed by testing the most abundant plants in between rows (*Medicago sativa*, *Trifolium repens*, *Cirsium arvense* and *Taraxacum officinale*), perennial plants in border areas (*Fraxinus americana*, *Ulmus americana*, *Rhamnus cathartica*) and wild grape (unknown *Vitis* sp.). The HTS result showed that cover crops and perennial plants did not harbor any grapevine viruses, while 4/5 wild grapes tested positive for GPGV but not GRBV. A pairwise sequence identity analysis revealed high similarities between the GPGV isolates found in the established vines on the vineyard and the newly contracted GPGV isolates in the sentinel vines, implicating a recent transmission event. This work provides novel insights into the spread of grapevine viruses in Niagara Region and is also the first direct proof of the spread of GPGV in natural vineyard conditions in North America.

## 1. Introduction

In recent years, the emergence of grapevine viral diseases is posing a serious threat to the profitability and sustainability of the grape and wine sector [1]. Amongst the 86 viruses discovered in grapevines, several can negatively affect the vine, ranging from reduced yield and stunted growth to vine death [2,3]. Two grapevine viruses that hold significant economic importance are grapevine red blotch virus (GRBV) and grapevine Pinot gris virus (GPGV).

GRBV is a single-stranded DNA virus whose genome consists of seven bidirectional open reading frames (ORFs) and is classified under the genus *Grablovirus* of the family *Geminiviridae* [4]. Within a few years after its discovery, the appearance of GRBV was reported in many regions around the globe in both red and white cultivars [5,6,7,8,9,10,11].

Transmission of GRBV can occur via grafting (primary) or via an insect vector (secondary) [5,12]. Thus far, the only confirmed insect vector for GRBV is *Spissistilus festinus*, commonly known as the alfalfa treehopper [4,13]. The rate of transmission by *S. festinus* was shown to be relatively low, both in a controlled greenhouse setting [4] as well as in a semi-controlled vineyard setting [13]. Other insect species such as *Colladonus reductus*, *Osbornellus borealis* and *Melanoliarus* spp. can GRBV carriers, but currently, their ability to transmit GRBV has yet to be confirmed [14]. *Vitis* species are the only known host of GRBV. These include commercial *Vitis vinifera* cultivars and rootstocks, wild grapes, interspecific hybrids, and table grapes [15,16]. Nevertheless, since only a limited number of plant species were tested for their ability to carry GRBV, the possibility of an alternative host for GRBV is still a looming threat.

GPGV is the causal agent of grapevine leaf mottling and deformation disease (GLMD) [17]. A GPGV-infected vine can demonstrate a wide range of symptoms, ranging from asymptomatic to severe leaf deformation and even vine death [18]. This disease has caused significant damage to the vineyards in Italy, especially in the Trentino region. However, in North America, the vast majority of vines detected with GPGV have shown mild or no symptom [19]. GPGV is a positive-sense, single-stranded RNA virus belonging to the genus *Trichovirus*, family *Betaflexiviridae*. The presence of GPGV has been confirmed in all five continents where grapevine is grown [20]. The only confirmed vector for GPGV thus far is the erineum mite (*Colomerus vitis*) [21]. Several studies from different grape-growing regions since then have shown divergent results, suggesting that the secondary spread of GPGV is dependent on the local environment and ecosystem [22]. In Canada, GPGV has also been documented in all four main grape growing provinces, namely Ontario (ON), British Columbia (BC), Quebec (QC) [19] and Nova Scotia (NS) [23]. Of the three provinces that conducted surveys for GPGV, ON has the highest relative incidence rate and by a large margin (34.3% for ON, 9.8% for BC [19] and 4.6% for NS [23]).

Because the spreading of a virus depends heavily on the local insect population [21,24], vineyards of different agronomic practices may demonstrate drastically different viral spreading patterns, even if they are in the same region and experience similar climates. The use of synthetic pesticides, herbicide and fertilizer have profound impact on the insect population [25,26], and thus, may affect the transmission of grapevine viruses as well. Unlike the conventional vineyards, certified organic vineyards in Ontario, Canada are prohibited from using synthetic products and instead, rely on naturally derived products (Ontario Ministry of Agriculture, Food and Rural Affairs—OMAFRA). Even though there have been extensive studies on yield, sustainability, environmental and economic implications of organic agriculture [26,27,28,29], the impact of this practice on the spreading of plant viruses, especially grapevine viruses, is still limited.

Sentinel plants are an effective tool to monitor the movement of a pest, including plant viruses [30,31]. The movement of the virus can be monitored by introducing virus-free plants to a virus-infected surrounding and comparing the viral status pre- and post-introduction. Detection of any newly contracted virus in the sentinel plants is a strong indication that the said virus is being spread by an insect vector.

Considering the threat that GRBV and GPGV pose to the Canadian wine industry, the main objective of this study was to investigate if secondary transmission of GRBV and/or GPGV occurs in Niagara Region, a major grape-growing region of Canada. The possibility of an alternative host in the local flora was also examined by testing the most abundant plant species present.

## 2. Materials and Methods

### 2.1. Sites Selection

Two vineyards in Niagara Region, henceforth referred to as vineyard A and B, were selected in this study due to the high prevalence of grapevine red blotch virus (GRBV) and grapevine pinot gris virus (GPGV) (W. McFadden-Smith. 2021. CGCN Webinar Series. Available at: https://www.cgcn-rccv.ca/site/webinars-cgcn [Accessed 14 May 2023]). Vineyard A is a certified organic vineyard. The selected site in vineyard A was a block of French hybrid, cv. Vidal blanc, located at the northwest corner of the vineyard, previously reported to have at least 80% incidence of GRBV and GPGV (data published in the webinar cited above). Approximately 10 m to the north and to the west of this block was a thin strip of unmanaged riparian area, populated by predominantly white ash (*Fraxinus americana*), elm (*Ulmus americana*) and buckthorn (*Rhamnus cathartica*). Wild grape (unknown *Vitis* sp.) was also found in this area, albeit less frequently than the other species. In the conventional vineyard B, the selected block was also a *cv. Vidal Blanc* block, located close to the middle of the vineyard and no non-grape perennial plant was found in close proximity to the sampling area (data published in the webinar cited above). This block was reported to have at least 85% incidence rate of GRBV and GPGV. The overall layouts of each vineyard can be found in Figure 1. To confirm the relative incidence rate of GRBV and GPGV, 20 vines from each vineyard were sampled and tested by PCR in January 2023. Each sample contains four leaves with intact petioles. The four leaves were collected from all four cardinal directions to minimize the impact of uneven viral distribution in the vine. After the PCR test, the first four samples from each vineyard were selected for HTS.

### 2.2. Sentinel Vine Material

Forty grafted *Vitis vinifera* cv. Chardonnay vines were obtained from Sunridge Nurseries Inc, a certified-virus-free nursery (Bakersfield, CA, USA). These vines were kept in the phytotron at Brock University for 2 years prior to this study. The phytotron conditions were 26 °C in the day and 15 °C at night, 15 h of combined sunlight and artificial light, humidity 60–80%. The 40 sentinel vines were sampled once for virus testing prior to being introduced to the vineyards on 1 July 2021, in the same protocol described above. Since the sentinel vines were expected to be free of economically important viruses, the 40 pre-introduction sentinel vine samples were combined into eight composite samples, each comprising five sentinel vines and were named CSV-1 to CSV-8 and tested with HTS. To confirm the HTS result, PCR tests for GRBV and GPGV were also conducted for these samples.

### 2.3. Sentinel Vine Deployment

Sentinel vines were introduced to the two targeted vineyards on 1 July 2021, in 5 litre pots filled with mycorrhizae soil (BX Mycorrhizae, ProMix, Hamilton, ON, Canada) and watered twice a week. They were placed in the gap under the trellis of the commercial vines, in two adjacent rows in an alternating manner (i.e., one sentinel vine per one commercial vine). The distance between two sentinel vines in the same row was approximately two to three meters, and the rows were three meters (10 feet) apart. The sentinel vines were relocated back to the phytotron at Brock University on 31 October 2021. Three vines (vines 3 and 19 from vineyard A; and vine 9 from vineyard B) died during this period due to mechanical damage. The remaining 37 vines, prior to being replanted in the phytotron, underwent a complete foliar removal and extensive roots washes to ensure that no insect species was brought to the phytotron. In September 2022, 15 months post-introduction, the 37 sentinel vines were sampled again with the same procedure. The long inoculation time was to ensure that GRBV and GPGV, if contracted from the field, had enough time to increase in titre and spread systematically within the vine. These samples were tested with HTS and PCR for GRBV and GPGV on an individual basis.

### 2.4. Potential Alternative Hosts for GRBV and GPGV

A survey was conducted at each vineyard to determine the three most abundant plant species on the vineyard floor (henceforth referred to as floor-covering plants). In vineyard A, they were alfalfa (*Medicago sativa*), white clover (*Trifolium repens*) and creeping thistle (*Cirsium arvense*). In vineyard B, they were creeping thistle, white clover and common dandelion (*Taraxacum officinale*). One composite sample containing five individual samples for each species from each vineyard was collected. For these species, four mature leaves facing four different directions were collected as an individual sample. Due to the proximity of the unmanaged riparian area to the study site on vineyard A, perennial plants and wild grape were also considered potential viral reservoirs. One composite sample for each of the three most abundant species (white ash, elm and buckthorn) were collected. For these samples, four mature leaves on four branches facing different directions were collected as an individual sample. Since wild grape was previously shown to be a host for GRBV [15], five individual wild grape samples were collected in the same protocol used in Section 2.1. The composite samples from floor-covering plants and perennial plants as well as the individual samples from wild grape were tested for GRBV and GPGV with both PCR and HTS.

### 2.5. Sample Handling, Nucleic Acids Extraction and PCR Procedure

Within 24 h after collection, each collected sample, whether individual or composite, was ground in liquid nitrogen with a sanitised ceramic molar and pestle. The powder was then divided into four aliquots of 500 mg and stored at −80 °C within four hours after pulverization. One aliquot was used for total RNAs extraction with Spectrum™ Plant Total RNA Kit (Oakville, ON, Canada). The extracted total RNAs then underwent PCR for GRBV and RT-PCR for GPGV. Quality control was done with Nanodrop 1C (Thermo Fisher, Mississauga, ON, Canada) and the cut-off threshold was set to be between 1.6 and 2.2 A260/A280 ratio.

For GRBV, we used the primer set GVGF1 (5′-CTCGTCGCATTTGTAAGA-3′) and GVGR1 (5′-ACTGACAAGGCCTACTACG-3′) described in Al Rwahnih, 2013 [32]. For GPGV, the primer set GPGV-6474F (5′-TTCTGGTGATCCAATGGTAAAGA-3′) and GPGV-6912R (5′-ATTGCAAAGGCCGCACACACTTG-3′) described in F. Moran, 2018 was used [33]. The thermal-cycling conditions of the two viruses were almost identical. Using the Bio-Rad thermocycler (Bio-Rad, Mississauga, ON, Canada), the thermocycling program was set as follows: initial denaturation (94 °C for 5 min), 34 cycles of denaturation (94 °C for 30 s), annealing (54 °C for 45 s) and extension (72 °C for 60 s), final extension (72 °C for 10 min) and termination (4 °C indefinitely). The PCR test for GPGV has an additional reverse transcriptions step (56 °C for 60 min) with SuperScript II (Thermo Fisher, Mississauga, ON, Canada) before the initial denaturation while the test for GRBV does not.

Another aliquot was used for the HTS pipeline. To enrich the viral titre in samples, a scaled-down version of the dsRNA extraction protocol (using 500 mg instead of 2 g starting material) described in Tzanetakis and Martin (2008) was used. Nanodrop 1C (Thermo Fisher, Mississauga, ON, Canada) was used for quality control. Samples with a A260/A280 ratio outside the range of 1.5 and 2.2 were reextracted. Concentrations of these samples were determined using Qubit 4, high-sensitivity RNA kit (Thermo Fisher, Mississauga, ON, Canada).

### 2.6. HTS Procedure

Extracted dsRNA [34] with sufficient quality (A260/A280 between 1.6 and 2.2) and concentration (at least 50 ng/μL or higher) underwent cDNA libraries preparation with TruSeq^®^ Stranded mRNA Library Prep Plant (96 samples) and Truseq RNA CD index plate (96 indexes) (Illumina, San Diego, CA, USA) following the manufacturer protocol. The prepared libraries were sequenced with MiSeq platform (Illumina, San Diego, CA, USA).

### 2.7. HTS Data Analysis

Virtool, a cloud-based bioinformatic pipeline developed by Dr. Mike Rott and CFIA, was used to scan for all viruses and viroid presence in each sample (https://www.virtool.ca/, accessed on 15 March 2023). In Virtool’s pipeline, adapter sequences and ambiguous regions are trimmed off by Skewer 0.0.2 and then subjected to FastQC 0.11.5 for quality assessment [35]. Reads with sufficient quality were scanned with PathoScope (a program based on Bowtie 2.2.2.3) for all plant viruses and viroids with a customized database derived from Genbank [35]. Viruses or viroids with more than 15%coverage or bearing more than 1000 matching reads were considered positive [35].

### 2.8. Sequence Identity Analysis

No GRBV was detected in the sentinel vines by either HTS or PCR. The only newly contracted grapevine virus detected in the sentinel vines was GPGV. To determine the relationship among GPGV isolates found in the sentinel vines, commercial vines and wild grape, a sequence identity analysis was conducted. HTS-generated reads from all samples that were found to be positive with GPGV with HTS in the previous section were imported to CLC Genomic Workbench 20.0.4 (CGW). They were then trimmed off with the threshold of Q20 (automatic Illumina’s adapters detection, maximum ambiguity was 2, retain homopolymers if present at both ends, with minimum read lengths of 20 and a maximum read length of 151). We first attempted to extract GPGV genomes from each sample using the reference assembly on “Grapevine Pinot gris virus, complete genome” (NCBI accession number: NC_015782.1). However, because none of the samples contained a full GPGV genome, we opted to construct a consensus sequence of the first open reading frame (ORF-1) instead. Starting at nucleotide 95th and ending at nucleotide 5632nd of the reference genome (5538 bp in length), ORF-1 is the longest ORF on the GPGV genome and contains the *replicase-associated protein* genes. Eight complete ORF-1s were recovered from the 35 GPGV-positive samples. The wild grape sample WG-1 contained a near-complete ORF-1, with an ambiguous segment of nine nucleotides toward the 3′-end (nucleotide 5232nd to nucleotide 5240th on alignment). However, considering that this segment was relatively short in comparison to the ORF-1 (9/5538 nucleotides or 0.16%), WG-1 was also included in this analysis. To better visualize the relationship among the GPGV isolates found in this study with GPGV isolates from other regions of the world, six additional ORF-1s from published sequences on NCBI were also included in the sequence identity analysis. These additional sequences were BC-1 and ON-1 from Canada (KU194413 and OK558797), S148 and S149 from the USA (MK514531 and MK514532) and fvgIs-12 and fvgIs-13 from Italy (MH087443 and MH087444). The total data pool of 14 ORF-1s were imported to Sequence Demarcation Tool version 1.2 (SDTv1.2), where they were aligned using the MUSCLE algorithm. The option to clusters isolates together via a neighbor joining tree was rejected for better visualization. Their pairwise distances were calculated using default parameters. A time-tree was constructed by first building a maximum likelihood model (an unrooted, ML tree with General Time Reversible model bootstraps value of 1000, branches with less than 70% bootstraps consensus were collapsed). From that maximum likelihood model, the time-tree was constructed using RelTime method by Takamura et al., 2012 and choosing apple chlorotic leaf spot virus (ACLSV), which belongs to the same Trichovirus genus as GPGV, as the outgroup. ACLSV was chosen because the accuracy of the time-tree is improved if the phylogenetic distance between the ingroups and outgroup is not too large [36]. No additional parameter was set.

## 3. Results

### 3.1. Viral Status of the Pre-Introduction Sentinel Vines with PCR and HTS

Virtool’s Pathoscope detected only two grapevine viruses and a viroid in the eight composite samples from pre-introduction 40 sentinel vines, namely GRSPaV, GTLaV and HSVd. GRSPaV and HSVd are considered ubiquitous in grapevine, with minor or no effect on the vine’s physiology [36,37]. The economic importance of GaTLV, if any, is currently unclear. GRSPaV and HSVd were detected from all composite samples (8/8) with high coverage (>80%) of GRSPaV and HSVd in 7/8 samples. GaTLV was also detected in 4 composites sample with significantly poorer coverage. The only sample with low coverage of GRSPaV and HSVd, sample CSV-6, also has the highest coverage of GaTLV. The follow-up PCR confirmed the absence of GRBV and GPGV in all sentinel vine pre-introduction (Table 1 and Appendix A).

### 3.2. Viral Status of the Post-Introduction Sentinel Vines with PCR and HTS

Viruses that were present in the pre-introduction sentinel vines, namely GRSPaV, HSVd and GaTLV were also found in the sentinel vines post-introduction. GRSPaV was detected in 26/37 samples and high coverage (>80%) of the genome was obtained in 11 samples. Similarly, HSVd was detected in 15/37 samples and high coverage was obtained in 12 samples. For GaTLV, 21/37 samples were found positive, with high coverage obtained in seven samples. Overall, the viral status of the sentinel vines pre- and post-introduction did not change with regards to GRSPaV, HSVd and GaTLV.

No GRBV was detected in any of the 37 post-introduction sentinel vines (Table 2 and Appendix A). However, a high number of sentinel vines acquired GPGV. A total of 14/37 sentinel vines were found positive with GPGV. Intriguingly, 13/14 GPGV-infected sentinel vines were the ones planted in vineyard A and only one GPGV-infected sentinel vine was from vineyard B (Table 2). The follow-up PCR test (Appendix A) confirmed the presence of GPGV in 12/14 of these vines, with the exceptions being samples SSV-02 and SSV-13, which could be attributed to the low virus titre in the sample, reflected in the poor genome coverage in the HTS results (Table 2).

### 3.3. Viral Status of Established Vines and Wild Grapes

The endpoint PCR confirmed that the two vineyards targeted in this study have high incidence rates of both GRBV and GPGV. Regarding GRBV, 19/20 (95%) vines from vineyard A and 13/20 (65%) vines from vineyard B tested positive. Similarly, 20/20 (100%) and 14/20 (70%) vines from vineyard A and B, respectively, tested positive for GPGV (Table 3). The gel-image of the PCR test for these samples can be found in Appendix A.

In the eight commercial vines selected for HTS, seven were found positive with GPGV and five were found positive with GRBV. We also found 100% incidence rate of grapevine Rupestris stem pitting-associated virus (GRSPaV) (8/8) and 75% incidence rate of hop stunt viroid (HSVd) (75%) (Table 2). Grapevine Tymo-like associated virus (GTLaV) was detected twice while grapevine leafroll 2 and 3 (GLRaV-2, GLRaV-3), grapevine red globe virus (GRGV), grapevine Syrah virus (GSyV) and grapevine virus B (GVB) were all detected once. For the five wild grape samples from vineyard A, all five samples were found positive with GRSPaV while four of them were found positive with GPGV. No wild grape samples were found positive for GRBV (Table 4).

### 3.4. Viral Status of Floor-Covering and Perennial Plants with HTS

The virus scan with Virtool did not detect grapevine viruses from any composite samples of floor covering or perennial plants. White clovers from both vineyards hosted a wide variety of viruses and are similar in overall viral status. White clover mosaic virus (WCIMV) and white clover cryptic virus 1 (WCCV1) were detected in alfalfa. Soybean Putnam virus (SPuV) was detected in creeping thistle from both vineyards, while tomato ringspot virus (ToRSV) was detected in buckthorn and white ash from vineyard A. No virus was detected in elm or dandelion. The follow-up PCR also confirmed the absence of GRBV and GPGV from these samples.

### 3.5. Phylogenetic Analysis of the Newly Acquired Virus in Sentinel Vines

The only newly acquired virus found in the sentinel vines in this study was GPGV. With HTS, GPGV was found in 25 samples, of which 14 were sentinel vines post-introduction, seven were established vines and four were wild grapes. In those 25 samples, the reference assembles yielded a complete ORF-1 of GPGV for eight samples. Six of them were sentinel vines (SSV-4, -5, -9, -16, -18, -41) and two were established vines (VA-1 and VA-4). The wild grape sample WG-1, which contained a near-complete OFR-1 with an ambiguous segment of nine nucleotides toward the 3′-end, was also included in this analysis. These nine sequences can be found on NCBI (accession numbers: OQ945219 to OQ945227)

The ORF-1 of GPGV isolates found in sentinel and established vines were almost identical, with a pairwise identity of 99% or higher. They also shared very high similarity with GPGV isolates from other locations in Canada and the USA, with pairwise identity in the range of 97% to 100%. In comparison, the Italian GPGV isolates were found to be more genetically distant, as their ORF-1 only shared 95% to 97% sequence identity with the GPGV isolates found in sentinel vines and established vines. Interestingly, the GPGV isolate from wild grape differed even more significantly from the Italian GPGV isolates, sharing only 93% sequence identity. The GPGV isolate in wild grape shared high similarity (97–99%) with the GPGV isolates found in sentinel, established vines and isolate ON-1 (another GPGV isolate found established vine in Ontario, Canada), and slightly less similarity to GPGV isolates from the USA (S148, S149 39) and BC, Canada (BC-1) with around 96% pairwise identity (Figure 2).

## 4. Discussion

The primary objective of this study was to investigate if GPGV and GRBV were being transmitted in vineyards in Niagara Region. Forty sentinel vines that were free of GRBV and GPGV were deployed to two vineyards with high-incident rate of both GRBV and GPGV. One vineyard is organic (vineyard A) while the other is conventional (vineyard B). Aside from the sentinel vines, samples from the present commercial vines, floor-covering plants, nearby perennial plants and wild grapes were also collected and tested for GRBV and GPGV using both PCR and HTS.

Our results confirmed the high prevalence of GPGV and GRBV in the commercial vines. Using HTS as a viral scanning tool, we also detected high prevalence of ubiquitous virus and viroid GRSPaV and HSVd in the commercial vines, with an occasional presence of other grapevine viruses. GaTLV was detected in two samples, while GLRaV-2, GLRaV-3, GRGV, GSyV and GVB were all detected once. Since GLRaV-2 and -3 are both economically important viruses, it would be beneficial to monitor the spread of these viruses in these vineyards for following seasons.

Regarding the viral status of non-*Vitis* species in and around these vineyards, the HTS results show a complete absence of grapevine viruses and viroids in the selected floor covering and perennial plants from both vineyards (Table 5). The absence of GRBV from these samples agreed with a study published in 2016, in which six herbaceous plant species (*Cynodone dactlyon*, *Eschscholzia californica*, *Kickxia elantine*, *Malva neglecta*, *Plantago lanceolate, Tribulus terresis*) and seven woody plant species (*Rosa* sp., *Rubus armeniacus*, *Heteromeles arbutifolia*, *Ilex* sp., *Quercus lobate*, *Quercus suber* and *Vitis californica x Vitis vinifera*) were tested and none were found to be true host for GRBV, except for *Vitis californica x Vitis vinifera* [15]. The absence of GRBV in our alfalfa sample is also consistent with previous studies which found that alfalfa is a non-host for GRBV [13,24]. A recent study in Hungary detected GPGV in ash (*Fraxinus* sps.) [42], which may seem to disagree with our result. However, since ash is by no means an obligatory host of GPGV, this contradiction could be a coincidence. Beside the expected results, such as the detection of WCIV and WCCV1 in white clovers from both vineyards, there were also a few unexpected incidences such as the detection of soybean Putnam virus (SPuV) in creeping thistle or traces of tomato ringspot virus (TRSV) in buckthorn and elm (Table 5).

In the riparian area of vineyard A, wild grape was also present. The HTS viral scan for five wild grape samples showed 100% infection of GRSPaV. None of the wild grape was found positive with GRBV, while four over five wild grape samples were positive for GPGV (Table 3 and Table 4). Even though the reported GPGV incidence rate in wild grape seems high (80%), the small sample size of only five vines prevented us from drawing any concrete conclusion. Regardless, this result shows that wild grape is a potential alternative reservoir for grapevine viruses. Testing for commercially important viruses in wild grape surrounding a vineyard could hence become beneficial, or even necessary, for regions with high viral pressure.

With the viral landscape of each vineyard established (commercial vines heavily infected with GRBV and GPGV, wild grapes have GPGV but not GRBV and other plant species are free of grapevine viruses), twenty GRBV and GPGV-free sentinel vines were deployed to each vineyard (a total of 40 vines). After four months of exposure followed by 11 months of inoculation in the phytotron, the HTS result revealed that none of the sentinel vines contracted GRBV. Regardless, to conclude that GRBV is not being transmitted in these two vineyards is premature at this point. The transmission of GRBV via an insect vector was previously found to be relatively slow. Even in greenhouse conditions with carefully designed arrangement to facilitate the transmission of GRBV, the rate of infection was quite low [13,24]. Considering that the sentinel vines in this study were exposed to infected vineyards without any transmission-facilitating arrangement for only four months, the sentinel vines simply may not have had enough opportunities to contract GRBV.

Surprisingly, a large proportion of the sentinel vines (14/37) was found positive with GPGV post-introduction (Table 2). None of the GPGV-infected sentinel vine displayed any discernable symptoms, concurring with previous studies that found the majority of GPGV isolates in North America to be asymptomatic [19,43]. A closer look at the 14 vines that contracted GPGV revealed that most of them (13/14) were the vines exposed to vineyard A while only one vine was from vineyard B. Considering that the GPGV incidence rate of the two vineyards were relatively close (100% for vineyard A and 70% for vineyard B), the striking different in infection rate is most likely due to differences in vineyard management. We hypothesize that Vineyard A, being an organic vineyard, has an elevated population of insects compared to the conventional vineyard B, leading to more feeding incidences and higher chances of virus transmission. However, since the details regarding the agronomic practices of each vineyard were not fully disclosed, our hypothesis remains speculative and would require further investigation before any concrete conclusion could be drawn.

The explosive spreading of GPGV in vineyard A was also very similar to another incidence documented in Southern France, in which 70% of GPGV-negative vines got infected within one year [22]. The wide spread of GPGV observed in this study may explain the high prevalence of GPGV (34.3%) previously reported in ON [19]. To ascertain that the GPGV strains found in the sentinel vines were the same strains found in commercial vines, a sequence identity analysis based on the ORF-1 was conducted. The results show that GPGV strains found in sentinel vines and commercial vines of both vineyards were remarkably close, sharing 99–100% sequence identity (Figure 2). The high-sequence similarity in tandem with the clear shift in viral status pre- and post-introduction has unequivocally shown that the sentinel vines had contracted GPGV from the commercial vines, making this study the first to document a direct transmission of GPGV in the vineyards of North America. With regards to the GPGV strains found in wild grapes, because only one ORF-1 was recovered from five samples, there is not enough information at this point to conclude whether wild grape shares the same GPGV strains with the commercial vines. Regardless, the sequence identity analysis shows that the GPGV isolate WG-1 is much closer to North American isolates (97–98% similarity) than the Italian isolates (93% similarity—Figure 2). This result is consistent with previous work conducted by Vu et al. [19], in which an unambiguous separation between North American GPGV isolates and European isolates was established.

To improve the confidence in our result, all sentinel vines underwent an endpoint PCR test for GPGV and GRBV. PCR and HTS results were in complete agreement, except for the GPGV status of three sentinel vines (SSV-2, SSV-13 and SSV-18 (Table 1 and Table 2). In previous studies, a false negative PCR result has been attributed to low virus titre, either due to insufficient inoculation time, uneven viral distribution or viral tire variation during the year [44,45]. For many grapevine viruses, PCR tests on cane samples proved to be more sensitive and robust [46]. However, in the present study, the sentinel vines were relatively young and not every vine produced enough cane to be tested. Thus, we were limited to leaf and petiole sample. Since HTS was shown to have superior sensitivity in numerous studies [35,47,48], we decided to accept the HTS result and consider the PCR result to be a false negative.

We also acknowledge that there are several limitations for this study. Firstly, the short exposure time of four months does not well represent the conditions that the perennial commercial vines endure. To effectively monitor the movement of plant viruses with low infection rate such as GRBV [13], a prolonged study of at least two years would be more effective. Secondly, the lack of information on the local insect population, especially the mite *Colomerus vitis*, prevent us from a potentially deeper understanding of the role of this insect in spreading GPGV. Hypotheses on GPGV’s vector other than *Colomerus vitis* have become more plausible since the discovery of a wide range of GPGV alternative hosts [42]. Lastly, because only a limited number of non-Vitis species were tested, the possibility of alternative hosts is not at all exhausted.

## 5. Conclusions

This work provided novel insights on the epidemiology of grapevine viruses in the vineyards in Niagara Region. All non-grape plant species subjected in this study were found free of grapevine viruses. Spread of GPGV was confirmed in both vineyards, serving as the first direct proof of secondary transmission or GPGV in North America. The rate of infection of GPGV in the organic vineyard was substantially higher than the conventional vineyard. Although we did not detect any GRBV in the sentinel vines, the previously reported slow rate of infection of GRBV necessitates longer-term studies.

## Figures and Tables

**Figure 1 biology-12-01279-f001:**
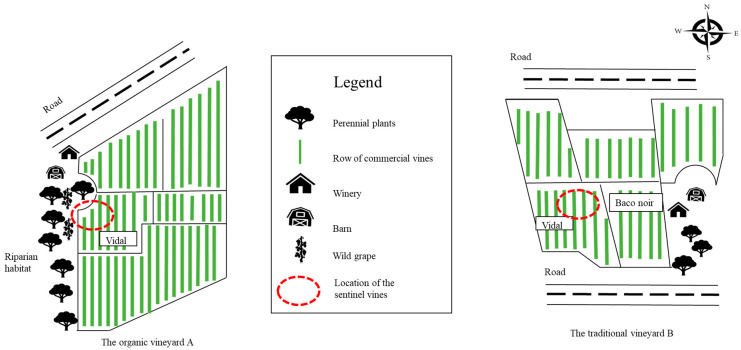
Overall layout of the two chosen vineyards in this study. The organic vineyard A is on the left and the conventional vineyard B is on the right. The compass on the upper right corner indicates the orientation of the vineyards. Note that these maps are a simplified depiction and are not to scale.

**Figure 2 biology-12-01279-f002:**
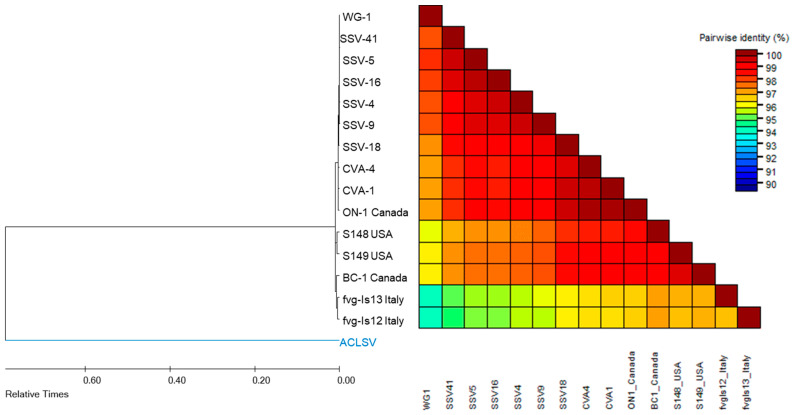
Phylogenetic tree inferring the evolutionary relationship and the color-coded matrix of pairwise identity scores among GPGV isolates from this study and a few representative GPGV isolates from Canada (BC-1 and ON-1), the USA (S148 and S149) and Italy (fvgIs12 and fvgIs13). The color-coded matrix was generated using the Sequence Demarcation Tool (SDT) v.1.2 (Muhire et al. 2014 [38]). The time-tree shown was generated using the RelTime (Tamura et al., 2012 [39]) method. Divergence times for all branching points in the topology were calculated using the Maximum Likelihood method and General Time Reversible model (Nei et al., 2000 [40]). The estimated log likelihood value of the topology shown is −18,125.22. A discrete Gamma distribution was used to model evolutionary rate differences among sites (five categories (+G, parameter = 0.9800)). The rate variation model allowed for some sites to be evolutionarily invariable ([+I], 10.84% sites). The tree is drawn to scale, with branch lengths measured in the relative number of substitutions per site. Evolutionary analyses were conducted in MEGA X (Kumar et al., 2018 [41]). The outgroup in this tree was ALCSV, a *Trichovirus*.

**Table 1 biology-12-01279-t001:** Viruses and viroid detected in the eight pre-introduction composite samples from the 40 sentinel vines detected with PathoScope-Virtool and their respective genome coverage. The PCR test result for GRBV and GPGV for these samples was also included.

Sample ^a^	HTS ^b^					PCR ^c^	
	GRSPaV ^d^	HSVd ^e^	GaTLV ^f^	GRBV ^g^	GPGV ^h^	GRBV	GPGV
**Vineyard A**							
CSV-1	100	100	27	0	0	-	-
CSV-2	100	100	0	0	0	-	-
CSV-3	100	100	0	0	0	-	-
CSV-4	94	100	15	0	0	-	-
**Vineyard B**							
CSV-5	98	100	0	0	0	-	-
CSV-6	39	73	41	0	0	-	-
CSV-7	100	100	23	0	0	-	-
CSV-8	100	100	0	0	0	-	-

^a^ CSV stands for composite sentinel vines, each composite sample represent five sentinel vine pre-introduction, ^b^ HTS—high-throughput sequencing, the number under the HTS column indicates the percentage of the genome cover for each virus, ranging from 0 (virus not detected) to 100 (total genome recovered). Viruses with genome coverage of 15% or more are considered present in the sample; ^c^ PCR—polymerase chain reaction. The negative sign (−) under the PCR columns indicates negative PCR test result while the positive sign (+) indicates positive PCR test result; ^d^ GRSPaV—grapevine Rupestris Stem pitting-associated virus; ^e^ HSVd—hop stunt viroid; ^f^ GaTLV—grapevine-associated Tymo-like virus; ^g^ GRBV—grapevine red blotch virus; ^h^ GPGV—grapevine Pinot gris virus.

**Table 2 biology-12-01279-t002:** Viruses and viroid detected in the post-introduction individual samples from the sentinel vines detected with PathoScope-Virtool, their respective genome coverage and the average depth. The PCR test results for GRBV and GPGV for these samples were also included.

Sample ^a^	HTS ^b^					PCR ^c^	
	GRSPaV ^d^	HSVd ^e^	GaTLV ^f^	GPGV ^g^	GRBV ^h^	GPGV	GRBV
**Vineyard A**							
SSV-01	80.9 (×9.85)	82.4 (×11.6)	0	0	0	−	−
SSV-02	26.4 (×14.39)	0	0	21.2 (×0.3)	0	−	−
SSV-04	99.9 (×15.75)	100 (×12.9)	83.4 (×3.4)	98.6 (×14.8)	0	+	−
SSV-05	99.9 (×7.18)	100 (×20.2)	61 (×3)	99.3 (×19.3)	0	+	−
SSV-06	97.7 (×17.94)	92.6 (×11.8)	78.9 (×10.1)	0	0	−	−
SSV-07	36.1 (×8.12)	0	30.5 (×1.8)	0	0	−	−
SSV-08	98.5 (×9.63)	100 (×14.7)	77.4 (×10.8)	79 (×6)	0	+	−
SSV-09	80.1 (×18.68)	0	82.6 (×8.4)	98.4 (×5.2)	0	+	−
SSV-10	24 (×17.67)	0	0	13.8 (×0.3)	0	−	−
SSV-11	99.7 (×5.99)	42.1 (×2.6)	0	81.6 (×13.1)	0	+	−
SSV-12	95.6 (×14.04)	0	30.8 (×1.2)	83.1 (×11.6)	0	+	−
SSV-13	43.1 (×8.08)	0	30.9 (×1.4)	37.4 (×2.7)	0	−	−
SSV-14	63 (×8.03)	0	29 (×0.5)	83 (×3.4)	0	+	−
SSV-15	34.7 (×13.05)	0	0	0	0	−	−
SSV-16	99.8 (×5.39)	100 (×5.26)	98.6 (×17)	99.4 (×20)	0	+	−
SSV-17	40.8 (×8.93)	50.8 (×2.1)	65.1 (×5.9)	96.7 (×6.4)	0	+	−
SSV-18	57 (×9.15)	100 (×6.7)	98.6 (×9.6)	99.8 (×10.8)	0	−	−
SSV-20	100 (×16.69)	100 (×19.6)	94.5 (×9.5)	99.1 (×5.4)		+	−
**Vineyard B**							
SSV-21	99.9 (×5.3)	100 (×9.4)	91.3 (×8.3)	99.4 (×13.1)		+	−
SSV-22	36.8 (×1.5)	100 (×9.7)	90.1 (×7)	0	0	−	−
SSV-23	0	0	0	0	0	−	−
SSV-24	27.1 (×1.5)	0	0	0	0	−	−
SSV-25	0	0	16.3 (×0.5)	0	0	−	−
SSV-26	28.5 (×1.7)	100 (×7.3)	20.5 (×0.4)	0	0	−	−
SSV-27	9.1 (0.1)	0	14.4 (×0.5)	0	0	−	−
SSV-28	1.7 (×0.1)	0	3.1 (×0.1)	0	0	−	−
SSV-30	15.8 (×0.3)	0	0	0	0	−	−
SSV-31	0	0	22.9 (×0.7)	0	0	−	−
SSV-32	0	0	11 (×0.3)	0	0	−	−
SSV-33	0	100 (×14.8)	46.7 (×3.3)	0	0	−	−
SSV-34	20.4 (×0.3)	0	0	0	0	−	−
SSV-35	0	0	0	0	0	−	−
SSV-36	10.5 (×0.2)	0	0	0	0	−	−
SSV-37	72.5 (×4.1)	0	21.3 (×0.7)	0	0	−	−
SSV-38	0	0	0	0	0	−	−
SSV-39	0	0	8.9 (×0.2)	0	0	−	−
SSV-40	99.9 (×5.3)	51.9 (×1.8)	0	0	0	−	−

^a^ SSV stands for single sentinel vines, each composite sample represent five sentinel vine pre-introduction; ^b^ HTS—high-throughput sequencing, the number under the HTS column indicates the percentage of the genome cover for each virus, ranging from 0 (no trace of the virus was detected) to 100 (total genome recovered). Viruses with genome coverage of 15% or more are considered positive. The number in the brackets next to the coverage is the average covering depth of that virus, calculated by the total number of nucleotides matched with the reference sequence divided by the total length of the reference sequence.; ^c^ PCR—polymerase chain reaction. The negative sign (−) under the PCR columns indicates negative PCR test result while the positive sign (+) indicates positive PCR test result; ^d^ GRSPaV—grapevine Rupestris Stem pitting-associated virus; ^e^ HSVd—hop stunt viroid; ^f^ GaTLV—grapevine-associated Tymo-like virus; ^g^ GPGV—grapevine Pinot gris virus; ^h^ GRBV—grapevine red blotch virus.

**Table 3 biology-12-01279-t003:** GRBV and GPGV status for 40 commercial vines and five wild grapes obtained with endpoint PCR.

**Vineyard A**					
**Sample ^a^**	**GRBV ^b^**	**GPGV ^c^**	**Sample**	**GRBV**	**GPGV**
VA-1	−	+	VA-11	+	+
VA-2	+	+	VA-12	+	+
VA-3	+	+	VA-13	+	+
VA-4	+	+	VA-14	+	+
VA-5	+	+	VA-15	+	+
VA-6	+	+	VA-16	+	+
VA-7	+	+	VA-17	+	+
VA-8	+	+	VA-18	+	+
VA-9	+	+	VA-19	+	+
VA-10	+	+	VA-20	+	+
WG1	−	+	WG2	−	+
WG3	−	+	WG4	−	+
WG5	−	+			
**Vineyard B**					
**Sample**	**GRBV**	**GPGV**	**Sample**	**GRBV**	**GPGV**
VB-1	−	+	VB-11	+	−
VB-2	+	+	VB-12	+	+
VB-3	+	−	VB-13	+	+
VB-4	+	+	VB-14	+	−
VB-5	+	+	VB-15	+	+
VB-6	+	−	VB-16	+	+
VB-7	+	−	VB-17	−	+
VB-8	+	+	VB-18	−	+
VB-9	+	+	VB-19	−	−
VB-10	+	+	VB-20	−	+

^a^ VA and VB stand for commercial vine from vineyard A and B, respectively; WG stands for wild grape; ^b^ GRBV stands for grapevine red blotch virus; ^c^ GPGV stands for grapevine Pinot gris virus.

**Table 4 biology-12-01279-t004:** Viruses and viroid detected in the eight individual commercial vines samples, five individual wild grape samples detected with PathoScope-Virtool, their respective genome coverage in percentage and the average depth. The number under each column indicates the percentage of the genome cover for each virus, ranging from 0 (virus not detected) to 100 (total genome recovered). Viruses with coverage of 15% or more are considered positive.

Sample ^a^	GRSPaV ^b^	HSVd ^c^	GPGV ^d^	GRBV ^e^	GaTLV ^f^	GLRaV-3 ^g^	GLRaV-2 ^h^	GRGV ^i^	GSyV ^j^	GVB ^k^
VA-1	99.9 (×15.2)	100 (×11.7)	91.3 (×6.3)	0	81 (×11.6)	0	0	0	0	0
VA-2	99.9 (×21)	100 (×12.8)	22.9 (×0.6)	33.9 (×0.8)	6.7 (×0.1)	0	0	29 (×1.5)	28 (×1.1)	0
VA-3	84.9 (×11.6)	100 (×11.8)	76 (×6.8)	16.3 (×0.3)	0	0	0	0	0	0
VA-4	79.4 (×8.5)	84.8 (×14.5)	94.6 (×11.4)	50.1 (×2.8)	0	0	0	8.6 (×0.1)	0	0
VB-1	100 (×20.5)	100 (×6.2)	48.6 (×2.8)	0	19.2 (×0.4)	97.0 (×14.2)	97.8 (×7.9)	0	0	30.8 (×1.1)
VB-2	99 (×14)	100 (×14.2)	31.1 (×2)	23.4 (×0.9)	81 (×11.6)	3.8 (×0.1)	0	0	0	0
VB-3	100 (×19.4)	100 (×11.7)	0	4 (×0.1)	6.7 (×0.1)	0	0	0	0	0
VB-4	95.1 (×7.3)	100 (×12.8)	33.1 (×1.6)	84.7 (×5.7)	0	0	0	0	0	0
WG-1	100 (×10.1)	0	96.7 (×14.1)	0	0	0	0	0	0	0
WG-2	99.9 (×4.5)	0	85.5 (×14)	0	19.2 (×0.4)	0	0.8 (×0.03)	0	0	0
WG-3	99.5 (×10.1)	0	35.7 (×1.5)	0	81 (×11.6)	0	0	0	0	0
WG-4	100 (×17.2)	0	82.8 (×10.6)	4.7 (×0.1)	6.7 (×0.2)	1.6 (×0.1)	0	0	10.8 (×0.2)	0
WG-5	70 (×3.2)	0	33.1 (×1.6)	0	0	0	0	0	0	0

^a^ VA and VB stand for commercial vine from vineyard A and B, respectively; WG stands for wild grape; ^b^ GRSPaV—grapevine Rupestris stem pitting-associated virus; ^c^ HSVd—hop stunt viroid; ^d^ GPGV—grapevine Pinot gris virus, ^e^ GRBV—grapevine red blotch virus; ^f^ GaTLV—grapevine-associated Tymo-like virus; ^g^ GLRaV-3—grapevine leafroll-associated virus 3; ^h^ GLRaV-2—grapevine leafroll-associated virus 2; ^i^ GRGV—grapevine red globe virus; ^j^ GSyV—grapevine syrah virus; ^k^ GBV—grapevine virus B. The number under each virus is the percentage of the genome cover for that virus, ranging from 0 (no trace of the virus was detected) to 100 (total genome recovered). Viruses with genome coverage of 15% or more are considered positive. The number in the brackets next to the coverage number is the average covering depth of that virus, calculated by the total number of nucleotides matched with the reference sequence divided by the total length of the reference sequence.

**Table 5 biology-12-01279-t005:** Plant viruses detected in floor covering and perennial plants in the proximity of the studied block. The number under the HTS column indicates the percentage of the genome cover for each virus, ranging from 0 (virus not detected) to 100 (total genome recovered). Viruses with more than 15% genome coverage are considered positive. The average depth is shown next to coverage.

**Vineyard A**										
**Category**	**Sample**	**WClV ^a^**	**WCCV1 ^b^**	**WCCV2 ^c^**	**CIYMV ^d^**	**CIYVV ^e^**	**RCVMV ^f^**	**White clover mottle virus**	**SPuV ^g^**	**ToRSV ^h^**
Floor-covering plants	Alfalfa	35.1 (×3.2)	24.5 (×1.1)	0	0	0	0	0	0	0
	White clover A	99.7 (×15.1)	96.2 (×9.7)	84.6 (×5.7)	83.8 (×7.2)	47.3 (×3.0)	19.6 (×0.9)	0	0	0
	Creeping thistle A	0	0	0	0	0	0	0	92.3 (×11.1)	0
Perennial plants	Buckthorn	0	0	0	0	0	0	0	0	39.7 (×1.6)
	White ash	0	0	0	0	0	0	0	0	48.5 (×2.2)
	Elm	0	0	0	0	0	0	0	0	0
**Vineyard B**										
**Category**	**Sample**	**WClMV**	**WCCV1**	**WCCV2**	**CYMV**	**CYVV**	**RCVMV**	**White clover mottle virus**	**SPuV**	**ToRSV**
Floor covering plants	White clover B	100 (×20.1)	90.0 (×9.2)	37.3 (×1.6)	65.2 (×4.1)	99.5 (×12.5)	99.5 (×16.4)	61.2 (×4.2)	0	0
	Creeping thistle B	0	0	0	0	0	0	0	96.8 (×7.2)	0
	Dandelion	0	0	0	0	0	0	0	0	0

^a^ WCIMV—white clover mosaic virus; ^b^ WCCV1—white clover cryptic virus 1; ^c^ WCCV2—white clover cryptic virus 2; ^d^ CIYMV—clover yellow mottle virus; ^e^ CIYVV—clover yellow vein virus; ^f^ RCVMV—red clover vein mosaic virus; ^g^ SpuV—soybean Putnam virus; ^h^ ToRSV—tomato ringspot virus. The number under each virus is the percentage of the genome cover for that virus, ranging from 0 (no trace of the virus was detected) to 100 (total genome recovered). Viruses with genome coverage of 15% or more are considered positive. The number in the brackets next to the coverage number is the average covering depth of that virus, calculated by the total number of nucleotides matched with the reference sequence divided by the total length of the reference sequence.

## Data Availability

The sequences used in the phylogenetic analysis of this study can be found on NCBI with the provided accession number.

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
