# Peer review of "Monitoring the Spread of Grapevine Viruses in Vineyards of Contrasting Agronomic Practices: A Metagenomic Investigation"

_biology, 2023, doi:10.3390/biology12101279_

Round 1
Reviewer 1 Report
The manuscript by Vu et al. describing the metagenomic investigation of monitoring the spread of grapevine viruses in vineyards is well written and the results are well discussed. The comparison of the spread of the virus under two different agronomic practices makes it more interesting. I have one suggestion for the authors:
1. As the authors have done several PCRs, the results (gel pictures) of these (at least the representative ones) should be included in the manuscript, while others can be attached as supplementary data.
The use of English language is okay.
Author Response
The first reviewer suggested that we add the gel-image of the PCR results to the paper. This is a great point. To implement this suggestion while avoiding the redundancy of data (repeated data between the tables and the gel-images), we decided to keep the images in a separate supplementary material file, which was submitted alongside this letter. This approach preserves the current flow of the article while providing sufficient data for interested readers.
Reviewer 2 Report
This manuscript “Monitoring the spread of grapevine viruses in vineyards of 2 contrasting agronomic practices: A metagenomic investigation” is a well executed study that investigates the secondary spread in two important grape infecting viruses using clean sentinel plants in vineyards with two management practices. They used a combination of HTS and PCR to determine the presence of these viruses. In addition, they looked at other possible wild reservoir plant species. This is clearly and concisely written and I only have minor suggestions and edits.

Author Response
The second reviewer suggested a few minor changes, all of which has been completed. Listed below are the suggestions from the reviewer and the changes that we made:
1) In the abstract: “especially in the organic vineyard” is a bit vague => We rewritten the sentence to “noticeably faster in the organic vineyard”.
2) In the abstract: specify “secondary infection” => changed to “insect mediated infection”
3) In 2.1 add reference to the infection rate => added reference
4) In Figure 1: fix that the borders do not overlap => fixed, also fixed text alignment
5) In 2.6: Map the raw reads on the GPGV and GRBV references => This procedure was completed and described in 2.7.
6) Throughout the document: OFR => ORF.
7) In Table 2: Add in average depth for each virus => Added, not just in table 2 but in all tables that contained HTS information
8) In Figure 2: Change the cladogram into a phylogram => Instead of the maximum-likelihood tree (a cladogram), we constructed a rooted time-tree (a phylogram) with relative time scale. This approach better illustrated the evolutionary relationship between the isolates and was a good addition to our paper.
Reviewer 3 Report
There are a number of viruses reported from grapevine. The authors claim that among them are two viruses of great economical importance: grapevine red blotch virus (GRBV) and grapevine Pinot gris virus (GPGV). GRBV is a grablovirus with a single-stranded circular DNA genome belonging to the family Geminiviridae, while GPGV is a positive-sense, single-stranded RNA trichovirus belonging to the Betaflexiviridae. The manuscript no biology-2544880 by Vu et al. reports on monitoring of the spread of GRBV and GPGV in two vineyards of conventional (B) and organic practice (A) in Canada. Sentinel vines originally free of the two viruses were introduced and grown for four months in either vineyard, and were then moved to and grown in environmentally controlled chambers for additional 11 months for virus detection. Note that previously established or existing vines in the two vineyards showed high incidence of the two viruses. The authors showed GPGV to be detected at a higher rate in the organic vineyard than in traditional vineyard B, whereas they could not detect GRBV from any of the sentinel vines. In order to search for possible reservoir plants of the two vine viruses, a variety of cover crops and perennial plants in the two vineyards were examined. However, none of the tested plants tested positive for either virus. Thus, the authors could not acquire any insight into virus reservoirs.
This reviewer feels this manuscript interesting as there is scarce information about transmission and dissemination of many grapevine viruses. However, there are two main problems that make this reviewer hesitant to recommend this paper for publication. First, this paper lacks sufficient amounts of biologically novel information. Second, this reviewer feels that this work would neither fall into the scope of the general biology journal Biology, nor attract its readership. The paper is more suitable for a specialized phytopathology or agronomy journal.
Good.
Author Response
The third reviewer stated that this paper might not be suitable for the overall theme of the Biology Journal. However, considering that this is a special issue focusing on “Advances in Plant Immunity against Viral Infection”, we believe our work on the dissemination of grapevine viruses and the first direct proof of insect-mediated viral infection of grapevine Pinot gris virus in North America may provide values to the readers of this special issue. Beside this comment, the third reviewer did not provide any other suggestion for improvement and thus, no adjustment was made.
Reviewer 4 Report
Vu and coll. report in this manuscript the results of a transmission study of two important grapevine viruses (grapevine red blotch virus, GRBV) and (grapevine Pinot gris virus, GPGV) in a conventional and an organic vineyard. To do so, they introduced a series of sentinel vines that have been previously tested and found free of these two viruses, and that were kept for 4 months in the vineyards, surrounded by plants infected in a very high percentage with these two viruses. After this exposure, the sentinel plants were transferred to a phytotron and kept there, where after 15 months the plants were tested for the presence of these viruses and others by HTS and (RT)-PCR. As a result, none of the sentinel vines tested positive by GRGV, while a high percentage of the sentinel vines that were introduced in the organic vineyard resulted infected by GPGV and only one in the conventional vineyard. A survey of perennial and herbaceous plants proximal to both vineyards showed the absence of these viruses in all these plants except for the wild vines proximal to the organic vineyard, which were found to be infected at a ratio of 4/5 by GPGV. The plausibly abundance of insect vectors in the organic vineyard probably explains the higher GPGV infection in the sentinel plants. This work is very interesting and provides relevant epidemiological information on the transmission of important grapevine viruses. It is very significant that in a period of only 4 months of exposure the plants were infected with GPGV.
Other comments: to help the reader, reducing the font size in the tables would make them more compact.
Author Response
The fourth reviewer suggested that the font size in the table should be smaller. We reduced the font of the characters in the table and compressed them to the best of our ability.
Round 2
Reviewer 3 Report
I thought that my review was appropriate and helpful to keep the standards and scope of the journal Biology. In response to my review, the authors claim that their manuscript was submitted to a special issue of the journal entitled “Advances in Plant Immunity against Viral Infection.” This reviewer still thinks that the paper does not fit into the SI, because of the lack of data that contribute to “Advances in Plant Immunity.” This reviewer wondered what part of the manuscript to was related plant immunity or immune responses. This reviewer is not defying the significance of the paper. Rather, the paper seems much more suitable for a phytopathology or agronomy journal. I hope that the Author and Editor understand this reviewer’s point.